# SMRS: advocating a unified reporting standard for surrogate models in the artificial intelligence era.

**Elizaveta Semenova**[*]
Imperial College London

**Alisa Sheinkman**
University of Edinburgh

**Timothy James Hitge**
AIMS South Africa

**Siobhan Mackenzie Hall**
University of Oxford

**Jon Cockayne**
University of Southampton

## Abstract

Surrogate models are widely used to approximate complex systems across science and engineering to reduce computational costs. Despite their widespread adoption, the field lacks standardisation across key stages of the modelling pipeline, including data sampling, model selection, evaluation, and downstream analysis. This fragmentation limits reproducibility and cross-domain utility – a challenge further exacerbated by the rapid proliferation of AI-driven surrogate models. We argue for the urgent need to establish a structured reporting standard, the Surrogate Model Reporting Standard (SMRS), that systematically captures essential design and evaluation choices while remaining agnostic to implementation specifics. By promoting a standardised yet flexible framework, we aim to improve the reliability of surrogate modelling, foster interdisciplinary knowledge transfer, and, as a result, accelerate scientific progress in the AI era.

## 1 Introduction

Surrogate modelling has become an indispensable tool for approximating complex computational systems across diverse fields, including engineering, natural sciences, machine learning, and artificial intelligence. By serving as cost-effective substitutes for expensive simulations or experiments, surrogate models (SMs) enable researchers to explore design spaces, optimise processes, and make predictions with significantly reduced computational overhead [1]. The landscape of surrogate modelling has evolved dramatically in recent years, driven in large part by advances in artificial intelligence. Traditional methodologies, such as nonparametric statistical methods (e.g., Gaussian processes [2, 3, 4], splines [5], hierarchical or mixed-variable models [6], and polynomial chaos expansion [7]) have been complemented, and in some cases outpaced, by modern AI-driven approaches. These include neural network-based surrogates [8], physics-informed neural networks [9], Bayesian neural networks [10], and conditional deep generative models [11]. Emerging paradigms such as world models, initially proposed in reinforcement learning to predict state transitions [12], and digital twins, originally defined as real-time digital mirrors of physical processes [13] but now broadly applied to system replicas, are further reshaping the field, pushing the boundaries of what surrogate models can achieve.

However, this rapid proliferation of AI-driven methods has led to a fragmented landscape. Application domains often rely on bespoke approaches, complicating performance evaluation and making it challenging to establish best practices. Key decisions, such as whether a model should be deterministic or probabilistic, whether it should incorporate domain knowledge, and how its quality should be assessed, are frequently handled inconsistently or overlooked entirely. These inconsistencies hinder

---

[*]Corresponding author: `elizaveta.p.semenova@gmail.com`

39th Conference on Neural Information Processing Systems (NeurIPS 2025) Position Paper Track.

reproducibility and cross-disciplinary knowledge transfer, limiting the broader impact of surrogate modelling advancements.

The primary objective of this paper is to make a case for such a **unified reporting standard**, particularly in light of the transformative impact of AI on surrogate modelling. A unified reporting standard would specify key components that each surrogate model should document, including input data formats, sampling design for data collection, model selection, training loss functions, evaluation metrics, uncertainty quantification, and performance on downstream tasks. By doing so, it would enhance the reliability of surrogate models and facilitate cross-domain collaborations, ensuring that advancements in one field can be effectively translated to others [14].

The need for standardisation is particularly pressing given the expanding use of SMs in critical applications such as climate modelling [15, 16, 17, 18], electric motor engineering [19], reservoir development [20], epidemiology [21, 22, 23, 24, 25, 26, 27], drug development, protein design [28], neuroscience applications [29], physical sciences [30], groundwater modelling [31, 32], wildfire nowcasting [33], and sea-ice modelling [34]. Without a common reporting standard, it is difficult to assess the reliability, limitations, and appropriate use of SMs, particularly as they are increasingly adopted in high-stakes domains. By proposing a structured and comprehensive schema for surrogate model reporting, this paper seeks to catalyse a shift toward more consistent and transparent practices in surrogate modelling. We emphasise that the Surrogate Model Reporting Standard (SMRS) we propose is intended to be lightweight and modular.

> 💡 **Position statement:** In the AI era, the field of surrogate modelling urgently needs a unified reporting standard that specifies the key components each SM should document, in order to address current inconsistencies, foster reproducibility, and enable cross-domain translation.

The remainder of the paper is organised as follows: Section 2 introduces the proposed unified reporting standard and its key components (with a link to the Case Studies demonstrating the evaluation of the proposed schema on several published surrogates); Section 3 presents alternative perspectives and addresses concerns about standardisation; and Section 4 summarises the contributions and offers a call to action for the community.

## 2 Proposed reporting schema for surrogate models

| Describe model to be emulated | Data collection | Model class | Learning methods | Evaluation methods |
|---|---|---|---|---|
| Motivate the choice of a surrogate model, describe (computational) gains and losses. | Describe source of the data, the sampling method, and use appropriate and standardised data formats. | Indicate and motivate the model class and structure, and describe underlying assumptions. | Document the chosen learning method, provide necessary details to foster reproducibility. | Document diagnostic and task-based evaluation, including relevant benchmarks. |

Figure 1: An Overview of the proposed Surrogate Model Reporting Standard (SMRS)

We propose a unified reporting schema (fig. 1) that identifies the essential components each application of surrogate modelling should document. These include data sources and formats, sampling design, model structure and assumptions, training methodology, evaluation criteria, and uncertainty quantification. Inspired by reporting practices such as "model cards" [35], this schema is adapted to the specific modelling practices, constraints, and scientific goals encountered in surrogate modelling across disciplines. Each stage of the modelling pipeline corresponds to a dimension along which developers are encouraged to explicitly report decisions made during the modelling process.

This structured approach is intended to promote transparency, improve reproducibility, and facilitate meaningful comparisons across surrogate models. In our review of 17 recent studies (see Case Studies), we observed substantial variation in which aspects of model development are reported. While most papers describe model architectures and predictive performance, few consistently report

how data were collected, how sampling decisions were made, whether uncertainty was quantified, or how limitations were handled. This heterogeneity makes it difficult to assess model reliability or applicability, particularly in high-stakes or cross-domain contexts.

The schema we present is modular and model-agnostic: it accommodates a broad range of modelling approaches, from classical statistical surrogates to modern AI-based generative methods. Figure 1 outlines the pipeline that motivates the reporting structure. The subsequent subsections detail each component, and the schema may be used as a checklist throughout the model development cycle: prior to, during, and after model training, to support methodological completeness and robust scientific communication.

## 2.1  Do you really need a surrogate?

Before addressing how to construct a surrogate model, it is important to first assess whether one is needed in the first place. Surrogate modelling refers to the *approximation of an existing, typically expensive, generative model* – often a mechanistic simulator or physical system. Unlike general predictive models trained on observational data, surrogates aim to *emulate* known processes that are slow, non-differentiable, or otherwise costly to evaluate. While all surrogate models are predictive, not all predictive models are surrogates. The defining feature is that a surrogate stands in for a known but costly or inaccessible process, rather than modelling observational data in the absence of a mechanistic ground truth.

A surrogate is typically appropriate when the goal is to replace a computationally intensive simulator or emulate a well-defined input–output relationship. If, instead, the aim is to predict from empirical data without reference to an underlying model, the task may better fit within standard supervised learning. In many cases, a surrogate may be unnecessary or even counterproductive. Performance issues may stem from inefficient code rather than model complexity, and can often be resolved via profiling and refactoring code, or using optimisation techniques such as vectorisation, parallelisation, or just-in-time compilation (e.g., using tools such as `JAX` [36] or `Julia` [37]). If existing surrogates suffice, or if the problem is low-dimensional and computationally cheap, additional modelling may be redundant. Likewise, highly non-linear or discontinuous systems may defy accurate approximation, favouring direct numerical or hybrid methods.

Surrogates are most useful when simulations are prohibitively expensive, the input–output map is sufficiently smooth, and repeated evaluations are needed; for example, in optimisation, design exploration, or uncertainty quantification. Clarifying these conditions early helps ensure surrogate modelling is applied where it offers real scientific or computational value.

> ⚙ **Motivating the choice to use a surrogate model, such as in the case of:**
> - ✔ Expensive data generation processes.
> - ✔ Input-output map has known, modellable structure.

## 2.2  Data collection and generation

The quality and representativeness of training data are critical to the success of surrogate models. In the following subsection, we outline different sampling designs to support experimental design. In the reporting mechanism, the researchers would document and motivate the sampling design choice to assist in reproducibility. If the training data do not adequately represent the system or process being emulated, the surrogate may exhibit biased behaviour, poor generalisation, or systematic prediction errors [38]. We recommend that surrogate model developers explicitly document how training data were obtained, the underlying assumptions, and the conditions under which the data are valid. This includes two core aspects: the mode of data generation and the sampling design. We also emphasise the importance of standardised data formats to support reproducibility and integration with domain-specific tooling.

**Two modes of input data.**  We distinguish two main modes of training data generation in surrogate modelling: (i) data collected from real-world systems and (ii) data generated from computational systems, including both offline ("pre-generated") and online ("on-the-fly") simulation.

*(i) Real-world data* are data acquired from physical experiments, observational campaigns, or operational systems such as sensor measurements in environmental monitoring, laboratory results in drug discovery, or clinical trial records in epidemiology. Real-world data are typically collected independently of the surrogate development process and are stored for later use. This mode introduces challenges around data quality, consistency, and completeness, especially when experimental conditions or metadata are poorly documented. Bias in measurement instrumentation or sample collection protocols may propagate into surrogate model errors if not accounted for explicitly.

*(ii) Computational data* are generated from numerical simulators, mechanistic models, or synthetic environments. They may be collected in two ways: an *offline* approach, in which simulations are run in batches and stored prior to surrogate training; or an *online* approach, in which new simulations are generated adaptively in response to the surrogate model's current uncertainty or performance. The online mode is particularly suited to active learning or Bayesian optimisation workflows. Recently, the use of AI-based simulators, including neural partial differential equation (PDE) solvers and physics-informed neural networks (PINNs), has enabled large-scale data generation at reduced computational cost, serving as a valuable foundation for training data-hungry surrogates.

**Sampling design.** Sampling strategy has a major influence on the representativeness, coverage, and generalisability of the surrogate model [39]. Simple strategies such as *random sampling*, which selects points uniformly at random from the input space, and *grid sampling*, which arranges points on a fixed mesh, are common starting points. While these approaches are straightforward to implement and interpret, they often fail to scale to high-dimensional spaces: random sampling can leave large gaps or clusters, while grid-based methods become infeasible due to the exponential growth in sample size with dimensionality.

An extension of random sampling involves drawing from a *non-uniform prior distribution*, which can be advantageous when prior knowledge is available about regions of interest in the input domain. For example, inputs known to be near decision boundaries or phase transitions may warrant denser sampling. However, in highly nonlinear systems, uniformity in the input space does not translate to uniformity in the output space, which can lead to biased surrogates unless this distortion is carefully accounted for.

Consequently, more sophisticated approaches follow *space-filling* or adaptive strategies. Common *space-filling* designs include *Latin hypercube sampling (LHS)*, *Sobol* and *Halton sequences*, and *orthogonal array*-based designs. These techniques offer better coverage with fewer points, especially in moderate-dimensional problems. For problems where certain regions of the input space are more informative or uncertain, *adaptive sampling* methods allow the model to iteratively propose new points for evaluation, guided by uncertainty estimates, error bounds, or domain knowledge. This helps balance the trade-off between exploration and exploitation.

A promising direction is to use *AI-based strategies* for adaptive exploration of the input space. Classical techniques such as *active learning (AL)* and *Bayesian optimisation (BO)* target regions of high uncertainty or expected gain, using query strategies that efficiently guide data collection [40, 41]. BO remains particularly efficient in low-dimensional spaces. Recent advances in *meta-learning* have further improved the rapid adaptation of sampling strategies across related tasks [42]. Extending these ideas, *reinforcement learning (RL)* offers a flexible framework, where a policy agent learns to propose input points that reduce surrogate model uncertainty or improve expected downstream performance [43, 44]. While RL offers dynamic adaptation, it often requires careful reward design and a substantial number of initial samples. Together, these approaches help mitigate the curse of dimensionality by focusing computational effort on informative regions of the input space.

**Standardised data formats.** To support transparency, reusability, and integration with domain workflows, we recommend using established data standards wherever possible. Domain-specific and general-purpose formats can improve interoperability across research groups and allow surrogate models to be evaluated or reused more systematically. Examples include NetCDF for climate science [45], SMILES strings in molecular chemistry [46], HDF5 for engineering [47], Parquet for large tabular datasets [48], and GeoTIFF for raster-based geographic inputs [49].

Metadata standards such as the CF Conventions [50] or Dublin Core [51] further support reproducibility by encoding information about units, spatial and temporal resolution, variable definitions, and provenance. Consistent formatting is especially important when applying *foundation-model-based*

data preprocessing or augmentation methods, which rely on predictable structure across inputs. When such formats are not used, authors should still clearly describe the structure, units, and assumptions encoded in their input data.

**Fidelity level.** Many scientific and engineering workflows provide outputs at several levels of fidelity: from coarse, inexpensive approximations to accurate but costly simulators. Multi-fidelity surrogates exploit this hierarchy by using low-fidelity data for broad exploration and high-fidelity data for refinement, thereby cutting the overall simulation budget [52, 53]. We therefore recommend documenting (i) which fidelity levels were used to train/evaluate the surrogate and (ii) how, if at all, cross-fidelity information was integrated (e.g., co-kriging, transfer learning, hierarchical loss terms).

**Heterogeneous or fidelity-ambiguous datasets.** In some scientific applications, surrogate models draw upon data from multiple simulators or measurement campaigns that differ in scale, noise level, or modelling assumptions, yet do not form a clear fidelity hierarchy. To accommodate such cases, the SMRS introduces a multi-source fidelity table in which each source is listed with its defining attributes, such as resolution, uncertainty level, governing equations or instruments, and mapped onto shared "fidelity axes" such as spatial, temporal, or physical realism. The comparability of each source can then be marked as comparable, partial, or not comparable. When no meaningful hierarchy exists, authors should indicate *N/A* and provide a short justification (e.g., "distinct experimental protocols, no common reference"). This structured record encourages transparent reconciliation of heterogeneous datasets and supports reproducible evaluation through stratified testing, such as leave-one-source-out cross-validation.

> ⚙ **Documenting the data collection pipeline**
>
> ✔ Modes of input data (real-world or computational data); collected online or offline.
>
> ✔ Motivation for the sampling strategy (random / grid / adaptive / AI-based strategies).
>
> ✔ Choose standardised data formats whenever possible and report the structure, units and assumptions encoded in any non-standard data formats.
>
> ✔ Fidelity level.
>
> ✔ Indicate data availability (otherwise, motivate its unavailability clearly, and include documentation to support use).

## 2.3   Model class selection

The choice of model class fundamentally shapes a surrogate's capabilities, including its expressiveness, scalability, and ability to represent uncertainty. We recommend that surrogate modelling studies explicitly report key properties of the selected model class: treatment of uncertainty, parameterisation structure, conditionality, and any domain-specific constraints. This section describes each of these components in turn. Critically, they should also report the underlying *model assumptions*, as these can strongly affect performance; for example, an inappropriate kernel choice in Gaussian process models can lead to catastrophic failure of the emulator.

**Deterministic vs probabilistic surrogates and uncertainty quantification.** *Deterministic* (point-estimate) surrogates, such as splines [5] or standard feed-forward neural networks, return a single prediction per input and are typically trained by minimising a scalar loss. They are computationally efficient and often adequate for systems governed by deterministic laws. In contrast, *probabilistic* surrogates output a full predictive distribution, which is indispensable when modelling inherently stochastic simulators (e.g. agent-based models [21]), when accounting for parameter uncertainty, or when downstream tasks demand uncertainty propagation. Gaussian processes, Bayesian neural networks, and deep generative surrogates such as diffusion models [34] exemplify this class. Where these techniques are too expensive, approximate probabilistic techniques such as variational inference [54] can be considered. Authors should state explicitly whether a surrogate is deterministic or probabilistic, and under what modelling assumptions. Deterministic surrogates can often be augmented to provide *uncertainty quantification* via add-on schemes, such as deep ensembles [55] or Monte-Carlo (MC) dropout [56, 57]. When surrogate predictions are used downstream, propagating epistemic and aleatoric uncertainty, or, failing that, performing a rigorous sensitivity analysis is essential to avoid biased decisions [58]. We therefore recommend *reporting* (i) whether predictive uncertainty is

estimated, (ii) the method used (ensemble, MC-dropout, VI, etc.), (iii) how the resulting uncertainty is applied in the modelling pipeline, and, if applicable, (iv) whether sensitivity analysis is performed.

**Parametric vs. nonparametric models.** *Parametric models* use a fixed set of parameters and a predefined functional form, making them suitable for large datasets and high-dimensional settings. Examples include classical neural networks, splines, and transformer-based surrogates [59, 60]. *Nonparametric models*, such as Gaussian processes, adapt their complexity to the data and are effective in low-data regimes or when the system is poorly characterised. We recommend specifying whether the surrogate is parametric or nonparametric, and what trade-offs this entails for scalability and inductive bias.

**Conditionality.** Many surrogates condition outputs not only on inputs but also on auxiliary variables, such as boundary conditions or environmental settings. These *conditional surrogates* are especially useful for scenario analysis or modelling system families [11]. We recommend reporting whether the surrogate is conditional and specifying the conditioning variables, as this impacts generalisability.

**Domain-specific constraints.** Surrogates often need to respect known constraints, such as physical laws, conservation, or monotonicity. These can be embedded in the architecture or enforced via the loss function. Examples include *PINNs* [9] for differential equations, *geometry-aware surrogates* [61], and *monotonic models* [62]. Incorporating such constraints enhances interpretability and robustness. We recommend stating whether and how domain constraints are applied.

---

> ✿ **Documentation guideline for the model class specifications. Indicate and motivate:**
> - ✔ whether the SM produces deterministic or probabilistic outputs, specifying conditions.
> - ✔ whether, and if applicable, how predictive uncertainty is used in the pipeline.
> - ✔ whether the model is parametric or nonparametric and the tradeoffs for scalability and inductive bias.
> - ✔ Conditioning variables, if applicable.
> - ✔ Domain-specific constraints (e.g. physical laws).

---

### 2.4 Learning methods

Surrogate models can be trained using a range of learning paradigms, depending on the surrogate's structure, the nature of available data, and the requirements of the downstream task. While optimisation remains the most common approach, alternative strategies such as Bayesian inference, simulation-based inference, and reinforcement learning are increasingly used, especially when uncertainty quantification or data scarcity are critical concerns. We recommend that surrogate modelling studies clearly report the learning paradigm used in enough detail to ensure reproducibility.

Most surrogates, whether deterministic or probabilistic, are trained by explicit *optimisation*: either using standard algorithms such as gradient descent or quasi-Newton methods [63] or stochastic-gradient methods such as Adam [64] or RMSProp [65], with meta-heuristics used when objectives are non-differentiable [66]. Classical *Bayesian surrogates* add a prior and compute the posterior explicitly in conjugate settings (e.g. with Gaussian processes) or via Markov chain Monte Carlo including its Hamiltonian variants [67, 68, 69]; when those are too slow, variational inference turns posterior estimation back into optimisation [54]. *Generalised Bayesian inference* [70, 71] widens this framework by replacing the likelihood with a task-specific loss,often a Bregman divergence [72] or proper scoring rule [73], to gain robustness to misspecification [74, 75]. High-dimensional or multi-modal outputs motivate *deep generative surrogates*, including variational auto-encoders [76], normalising flows [77, 78], diffusion models [79] and neural likelihood estimators [80, 81]. When explicit likelihoods are unavailable but simulators exist, simulation-based inference trains the same architectures on synthetic data [82, 83, 84]. Adaptive scenarios can instead be framed as *reinforcement learning* [85], in which a policy selects inputs or refines the surrogate to maximise an information-theoretic or task reward. Finally, *AutoML* systems automate architecture and hyper-parameter search through *Bayesian optimisation* or evolutionary strategies [86]. For any of these paradigms, authors should state: the objective or loss being optimised; the optimiser, sampler or RL algorithm; the learning-rate schedule and early-stopping rules; the prior or scoring rule if Bayesian or

generalised Bayesian; the design and volume of simulator runs if using simulation-based inference; and the search space and evaluation metric when employing AutoML.

> ⚙ **Document the chosen learning method and the applicable details**:
> - ✔ Optimisation-based learning: optimiser, training schedule, any regularisation and early stopping criteria.
> - ✔ Bayesian inference: inference algorithm and prior distributions.
> - ✔ Generalised Bayesian inference: the specific task-based loss function for inference.
> - ✔ Probabilistic surrogates with generative AI: generative architecture, objective function, method for sampling or posterior inference.
> - ✔ Simulation-based inference: procedure for generating simulations, summary statistics or features used, and conditioning on these.
> - ✔ RL and adaptive objectives: RL formulation (state, action, reward), learning algorithm, downstream objective being optimised.
> - ✔ AutoML for surrogate model selection: indicate how model selection was automated, alternatives considered and how stable the results are across initialisations.
> - ✔ Open source all code used to implement the chosen methods.

## 2.5 Evaluation metrics and benchmarks

A surrogate model must be evaluated not only for its predictive accuracy, but also for how well it supports the scientific or engineering task for which it was developed. We distinguish between *diagnostic evaluation*, which assesses statistical properties of the surrogate model itself, and *task-based evaluation*, which measures how well the surrogate performs in downstream applications. We recommend that all surrogate modelling studies report their evaluation methodology, including the metrics used, any ground truth comparisons, and details of benchmarking datasets or protocols.

While evaluation metrics often mirror the training objective, they can and should diverge to assess broader aspects such as uncertainty quantification, calibration, robustness, and domain-specific requirements. For example, a model trained via likelihood maximisation may be evaluated using calibration or task-level regret instead of predictive accuracy alone.

**Diagnostic evaluation: distributional similarity.** For *probabilistic* surrogate models, it is vital to measure how well the predicted output distribution aligns with that of the true simulator. Divergence metrics such as the *maximum mean discrepancy (MMD)* [87], the *Fréchet distance*, implemented as the Fréchet Inception Distance (FID) in generative tasks [88], and the *sliced Wasserstein distance* are effective in high-dimensional settings. These diagnostics are indispensable when the surrogate seeks to capture epistemic or latent variability. A recent survey [89] maps metric choice to data modality. Authors should state the metric used, its feature representation, and whether distances are computed per output dimension, jointly, or via summary statistics.

**Diagnostic evaluation: calibration.** Calibration gauges how well a model's stated uncertainty reflects the variability seen in data. For Bayesian and other *probabilistic* surrogates, *simulation-based calibration (SBC)* [90, 91, 92] validates posterior consistency by: (i) sampling synthetic datasets from the prior predictive, (ii) refitting the model to each dataset, and (iii) checking that ground-truth parameters rank uniformly within the recovered posteriors. A well-calibrated surrogate yields a flat rank histogram. We advise performing and reporting SBC or an equivalent diagnostic whenever surrogate predictions are probabilistic or used in downstream inference.

**Diagnostic evaluation: estimated generalisation performance of the model.** Measures such as *Akaike information criteria (AIC)* [93] and *Deviance information criteria (DIC)* [94] can be used to compare several statistical models and asses their predictive performances. In a fully Bayesian setting, leave-one-out cross-validation (LOO-CV) and the *Watanabe-Akaike information criterion (WAIC)* both provide a nearly unbiased estimate of the expected predictive ability of a given model [95] and can be used to compare and average several candidate models [96].

**Task-based evaluation.** In many applications, surrogate models are embedded within broader decision-making pipelines. Examples include *Bayesian optimisation* [41], *active learning* [97], *design of experiments* [98], and *uncertainty propagation* [99]. In these contexts, surrogate quality should be assessed based on how it affects performance in the downstream task. For example, in optimisation, *regret* quantifies the efficiency with which the surrogate identifies the optimum. In uncertainty quantification, metrics such as coverage probability or predictive intervals may be more informative than mean squared error. We recommend that evaluation reports include not only predictive metrics, but also task-aligned performance indicators.

**Task-based evaluation protocols and stress testing.** In addition to conventional accuracy measures, the SMRS encourages authors to report metrics that directly reflect performance on the scientific or operational task for which the surrogate is built. Each study should specify a task family (e.g., optimisation, uncertainty propagation, control, forecasting) and report at least two task-relevant metrics. Examples include: quantile or extreme-event errors for climate and risk modelling; calibration and coverage probabilities for probabilistic inference; regret or sample efficiency for optimisation; and computational cost or energy use for engineering applications. Authors should also provide (i) a simple baseline model for context, (ii) a statement of compute budget or wall-time used for evaluation, and (iii) at least one stress test that probes model robustness, such as out-of-distribution scenarios, long-horizon predictions, or noisy boundary conditions. All metrics, baselines, and test conditions should be reported alongside random seeds and code paths, enabling like-for-like comparison across papers. When certain metrics do not apply, the relevant entry in the SMRS card can be marked N/A with a short justification. This explicit structure ensures that task-specific evaluation becomes reproducible and comparable across domains.

**Benchmarking.** Unlike *task-based evaluation*, which inspects a single surrogate on one user-defined task, *benchmarking* pits multiple surrogates against a shared testbed, enabling fair comparison and reproducibility. Benchmark suites should span diverse modalities (e.g., PDE solvers, agent-based models, environmental simulators), include ground-truth outputs, and record the assumptions of both data and task. Ideally, they expose multi-fidelity levels and scenarios that probe distribution shift or domain extrapolation. Generic repositories such as OpenML, UCI, and Kaggle help, but surrogate-specific, versioned datasets with unified protocols remain scarce. We therefore call for *curated, trust-aware* benchmarks [100], and encourage authors to declare whether they used public benchmarks, in-house simulators, or synthetic data, and to release code or protocols. Adaptive, AI-generated test distributions may further stress-test surrogates and highlight failure modes.

> ⚙ **Extensively document both diagnostics and evaluation.**
> - ✔ Predictive performance: diagnostic evaluation such as calibration, distributional similarity, and estimated generalisation performance of the model.
> - ✔ Downstream task performance: task-based evaluation and benchmark (indicate if in-house, public, and/or synthetic datasets were used).

## 2.6 Case studies

To illustrate how existing research aligns with or diverges from our unified surrogate modelling framework, the Case Studies provides reviews of three recent studies using SMs. These papers were selected as examples after reviewing 17 papers from various fields and determining which had the most detail to demonstrate the implementation of the SMRS framework. Each study is mapped onto the key components of our pipeline. Overall, these case studies reveal both the versatility of surrogate modelling, evidenced by its diverse applications, and the fragmentation that occurs in practice. However, the potential to learn from and reproduce these SMs is difficult without a standardised reporting framework. Extracting the relevant information from dense articles in a different domain is difficult and places an immense burden on the researcher attempting to do so. We hope that the demonstration of the SMRS in practice will speak volumes and demonstrate the ease with which relevant information can be accessed.

# 3 Alternative views

While a unified reporting schema for surrogate models promises significant benefits, several alternative perspectives raise valid concerns about feasibility, scope, and potential unintended consequences. We address three key critiques below and explain how our framework is designed to accommodate these concerns.

**Domain-specific practices are essential.** A common concern is that surrogate modelling workflows are inherently domain-specific. For example, climate science demands rigorous uncertainty quantification [101], while aerospace applications prioritise real-time prediction and strict physical constraints [1]. Tools such as the Surrogate Modelling Toolbox (SMT) [6] and physics-informed neural networks (PINNs) [9] succeed precisely because they are tailored to specific physical regimes and data structures. Critics may argue that standardisation risks enforcing generic practices that are misaligned with field-specific constraints, such as molecular string representations in drug discovery [46] or stochasticity in agent-based epidemiological models [24]. However, our proposed schema is intentionally modular and non-prescriptive. It aims to harmonise reporting practices, such as how uncertainty is characterised or how sampling design is justified, without constraining modelling choices. Domain-specific formats, simulators, and evaluation protocols can be retained, while benefiting from improved clarity, reproducibility, and comparability. We view the framework as an enabler of cross-pollination, not a constraint on disciplinary depth.

**Integration with existing tools and incentives is challenging.** Many scientific and engineering fields rely on legacy software (e.g., `MODFLOW` for groundwater modelling [102], `NetLogo` for agent-based systems [103]) or proprietary environments (e.g., `ANSYS`[2], `COMSOL`[3]), which often use bespoke data formats and provide limited support for modern metadata or validation protocols. Retrofitting such systems for standardisation can be not only technically daunting but also misaligned with prevailing academic and industrial goals, which often prioritise rapid results and project-specific outputs over generalisation, benchmarking and tedious documentation. Given these challenges, we advocate for incremental and pragmatic adoption. For example, workflows based on pre-generated simulation data (Section 2) can integrate standardised reporting during post-processing, without altering upstream simulators. Experiences from efforts such as the FAIR principles [104] demonstrate that reporting practices can evolve when supported by suitable tools, community engagement, and institutional mandates.

**Standardisation may inhibit innovation.** A final concern is that any formal schema risks ossifying the field or prematurely constraining methodological innovation. Fast-evolving areas like generative surrogates (e.g., diffusion models [34]) or transformer-based architectures [8] often outpace benchmarking infrastructure or evaluation protocols. We emphasise that the proposed schema does not prescribe particular models, training paradigms, or data modalities. Rather, it encourages transparent documentation and consistent reporting across diverse approaches. This improves interpretability, enables reproducibility, and accelerates innovation by making differences in assumptions and performance explicit. Open-ended benchmarks and modular checklists allow new methods to be compared without suppressing novelty.

These critiques highlight legitimate tensions between flexibility and standardisation. Our framework responds by focusing on *reporting*, not regulation: it is designed to surface assumptions, highlight best practices, and support methodological rigour, not to dictate modelling choices.

# 4 Discussion

In this work, we have advocated for a unified reporting schema to address the growing methodological fragmentation in surrogate modelling. Our review (see Case Studies) highlights both the versatility of surrogate models and the lack of consistent documentation across domains and model classes. To address this gap, we propose the Surrogate Model Reporting Standard (SMRS), a lightweight, modular framework inspired by efforts such as model cards [35] and the FAIR principles [104].

---

[2]ANSYS is available at the following URL: `https://www.ansys.com/`

[3]COMSOL is available via the following URL: `https://www.comsol.com/`

Rather than prescribing specific tools or architectures, SMRS encourages transparent reporting of key modelling decisions, including data provenance, sampling design, model assumptions, learning method, and evaluation strategy. While many of these elements are already reported in practice, this is often done in an informal or inconsistent manner. SMRS provides a structured format aimed at improving reproducibility, supporting benchmarking, and enabling cross-disciplinary reuse. It is designed to be model-agnostic, domain-adaptable, and incrementally adoptable.

While many research papers now share code and data, *open repositories alone do not ensure transparency or reproducibility*. Codebases rarely state the scope, assumptions, or validity ranges of a surrogate model, and often omit design decisions that affect reuse. The Surrogate Model Reporting Standard complements open-sourcing by introducing a declarative layer that summarises *what was done* and, very importantly, *what was not done*. It explicitly records data source and generation process (e.g. offline / online generation, active-learning criteria), clarifies model assumptions (deterministic vs probabilistic, uncertainty calibration), and documents evaluation protocols and computational context (hardware, seeds, commit hashes). This concise, standardised record enables like-for-like comparisons across studies, improves auditability, and supports independent reproduction even when repositories evolve or become unavailable.

As surrogate models are increasingly deployed in high-stakes settings, from climate science to healthcare, standardised reporting will be essential for ensuring scientific credibility and real-world impact. We position SMRS as a pragmatic foundation for building more interpretable, reliable, and transferable surrogate modelling.

**Practical adoption pathways.** The SMRS is designed for incremental, low-friction integration into existing publication and review workflows. *For authors,* the recommended practice is to include a concise one-page SMRS card in the appendix and repository, listing all schema fields with evidence links, or marking Not reported / N/A where appropriate. This clarifies completeness and helps future researchers identify missing information. *For reviewers*, a condensed SMRS checklist offers a rapid way to verify methodological transparency and request clarifications before acceptance. The checklist focuses on coverage of the key components (data, model, learning, evaluation, and task performance), rather than on implementation details. *For benchmark and survey creators*, SMRS cards can serve as a foundation for curated, auditable repositories. Benchmark organisers may release third-party-audited SMRS cards for each dataset and model, enabling like-for-like comparison of surrogates under common metrics and protocols. By making reporting structured but not prescriptive, SMRS encourages community-driven convergence towards reproducible and interpretable surrogate modelling without introducing unnecessary administrative burden.

## Acknowledgements

E.S. acknowledges support in part by the AI2050 program at Schmidt Sciences (Grant [G-22-64476]). T.H. is a Google DeepMind scholar and acknowledges the African Institute for Mathematical Sciences (AIMS), South Africa, for the opportunity to conduct this research as a part of his MSc. J.C. is partially supported by EPSRC grant EP/Y001028/1.

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

# Case Studies

In the following section, we detail the implementation of SMRS in four exemplar papers [105, 106, 29, 107]. Before selecting these exemplar papers, we initially audited 17 publicly available papers from fields of climate science [108, 109, 105], complex systems [110], agent-based systems [111, 112], geophysical and Earth based modelling [113, 114, 106, 115, 116], mechanics [117], material science [118] automated experiments modelling [119], as well as general physics [107], engineering [120] and deep learning applications [121]. While these selected papers are comprehensive, some details are missing, which we determine to the best of our knowledge, bearing in mind that completing these relies on cross-discipline assessments. The burden of interpretation is another motivating factor for the authors to do this in conjunction with the surrogate development. We assess the extent of the reported details with a basic code system, where a ✔ indicates sufficient detail provided by the authors, a **?** indicates some details were missing, and a ✖ indicates the detail is not provided at all. In the perfect implementation of SMRS, all details would be included and marked with a ✔.

## Contents

# A  Using surrogate modelling to predict storm surge on evolving landscapes under climate change

---

## Using surrogate modelling to predict storm surge on evolving landscapes under climate change

**Link:** `https://www.nature.com/articles/s44304-024-00032-9`

**Application Field:** Coastal flood hazard research.

### Motivation for using a surrogate model

- ✔ **Computationally expensive simulator:** Physically-based simulators are expensive and storm surge risk assessments require a large number of synthetic storm events.
- ✘ **Computational gains and losses:** The paper reports that the full surrogate model can be trained in under 7 hours on an AMD Epyc 7662 CPU and that predictions for a novel landscape can be generated within 4 minutes. However, this performance is not compared against the expensive simulator.

### Data Collection

- ✔ **Modes of input data:** Computational data was collected offline by extracting peak storm surge elevations at 94,013 distinct locations per simulation from a coupled ADvanced CIRCulation (ADCIRC) and Simulating WAves Nearshore (SWAN) model; these simulations included 645 synthetic tropical cyclones on a 2020 baseline landscape, and the same subset of 90 synthetic storms on each of 10 future evolving landscapes representing decadal snapshots from 2030 to 2070 under two different environmental scenarios.
- ✔ **Sampling Strategy and Motivation:** Joint Probability Method with Optimal Sampling and heuristic algorithms to reduce the number of simulations.
- ✔ **Data Formats:** The dataset consists of landscape and storm parameters, with associated peak storm elevations. The landscape parameters for the locations are stored in GeoTIFF files, which have a resolution of 30 meters. These files detail the topography/bathymetry, Manning's n value, free surface roughness, and the surface canopy coefficient. The storm parameters are stored in a CSV file and characterise the tropical cyclones by their overall tracks and five parameters at landfall; forward velocity, radius of maximum windspeed, central pressure, landfall coordinates, and heading. The peak storm elevations for the locations are stored in NetCDF files and their values are in meters relative to the North American Vertical Datum of 1988 (NAVD88).
- ✔ **Fidelity Level:** The surrogate model is trained using data derived from a single, high-fidelity simulation source: the coupled ADCIRC+SWAN numerical model.
- ✔ **Data Availability:** Available at ADCIRC Simulations of Synthetic Tropical Cyclones Impacting Coastal Louisiana.

### Model Class

- ✔ **Deterministic or Probabilistic?:** Deterministic.
- ✔ **Predictive Uncertainty:** N/A.
- ✔ **Parametric Model:** Artificial Neural Network (ANN) with four hidden layers (128–256 neurons each), ReLU activation in hidden layers, and linear output layers.
- ✔ **Conditioning variables:** Storm parameters, landscape features (topography, bathymetry, vegetation), and boundary conditions (e.g., sea-level rise).
- ✔ **Domain-specific constraints:** Not incorporated.

### Optimisation-based Learning

- **?** **Optimiser and loss function:** Not reported or motivated in the paper. In the code made available, the Adam optimiser is used with a MSE loss.
- **?** **Training Schedule:** The paper reports that the *full model* was trained for 100 epochs but does not specify the number of epochs that the *storm-only models* were trained for. Within the provided code, the *storm-only model* is trained for 1000 epochs, whilst it is unclear how many epochs the full model uses by default. Learning rate was set to $1 \times 10^{-3}$.
- **?** **Regularisation and early stopping criteria:** Not reported or motivated in the paper. In the supplied code, the *storm-only models* are trained without dropout, whilst the *full model* is trained with dropout.
- **?** **Code availability and usability:** Available at Storm Surge Prediction Over Evolving Landscape. Minimal usage instructions and installation instructions are absent.

### Diagnostics and Evaluation

- ✔ **Predictive performance:** The value of the inclusion of landscape parameters for a predictive model of a single landscape was assessed by comparing predictive accuracy on 15% of the storms held out for testing purposes. The generalisation performance of the multi-landscape model was assessed with a LOOCV procedure. In each iteration of the procedure, the model was trained using data from the 2020 landscape along with 9 of the 10 future landscapes, and predictions were subsequently made on the future landscape that had been held out.
- ✔ **Downstream task performance:** Annual Exceedance Probability (AEP) distributions are crucial for coastal flood protection projects as they quantify the likelihood and severity of flood events. AEP distributions were generated using the simulated surge elevations from ADCIRC and from predicted surge elevations from the surrogate LOOCV procedure using the Coastal Louisiana Risk Assessment (CLARA) model. The resulting empirical distributions were compared using a two-sample Kolmogorov-Smirnov test.

## B  Equation-free surrogate modelling of geophysical flows at the intersection of machine learning and data assimilation

---

**Equation-Free Surrogate Modelling of Geophysical Flows at the Intersection of Machine Learning and Data Assimilation**

**Link:** `https://agupubs.onlinelibrary.wiley.com/doi/10.1029/2022MS0031 70`

**Application field:** Geophysical flow modelling where the surrogate model is used to emulate sea surface temperature (SST) dynamics via a data-driven reduced-order model.

### Motivation for using a surrogate model

✔ **Computationally expensive numerical methods:** Numerical methods require the use of a supercomputer.

✘ **Computational gains and losses:** The paper reports that the full surrogate model can be trained on a CPU on the order of a minute, with inference taking place on the order of tens of milliseconds. However, the authors only allude to the computational requirements of the numerical methods and do not provide a direct comparison.

### Data Collection

✔ **Modes of input data:** Real world recorded observational data, collected offline.

? **Sampling strategy and motivation:** Historical data is used primarily. The training data are taken from October 1981 to December 2000 (1,000 snapshots), and the data from January 2001 to June 2018 (914 snapshots) is used to test the forecasting methods (predictive inference). QR pivoting selects near-optimal sensor locations for partial observations. No motivations for these choices are given.

? **Data formats:** Global Sea Surface Temperature (SST) data on a $1°\times1°$ grid ($180\times360$), then reduced via Proper Orthogonal Decomposition (POD). It is unclear from the paper if this is the field standard.

✔ **Data Availability:** Data is documented and uploaded to the following URL: `https://psl.noaa.gov/data/gridded/data.noaa.oisst.v2.html`

### Model Class

✔ **Deterministic or probabilistic?:** Deterministic.

✔ **Predictive uncertainty:** Deterministic ensemble Kalman filter (DEnKF) is used to propagate uncertainty through the system over time. The initial ensemble is created by adding random perturbations to the initial state estimate.

✔ **Parametric and Non-parametric:** The surrogate consists of a non-parametric Proper Orthogonal Decomposition for dimensionality reduction of time series of spatial snapshots of SST data and a parametric Long Short-Term Memory (LSTM) neural network with three stacked layers, each containing 80 units and ReLU activation.

✔ **Conditioning variables:** The LSTM network predicts the next state of the POD modal coefficients based on a lookback window of the previous four POD coefficients.

✔ **Domain-specific constraints:** Not incorporated.

- ✔ **Optimiser and loss function:** Adam optimiser and MSE loss.
- ✔ **Training schedule:** Learning rate of $1 \times 10^{-3}$. The LSTM is trained for 200 epochs with a batch size of 32.
- ✘ **Regularisation and early stopping criteria:** Not indicated.
- ❓ **Code availability and usability:** The link to the GitHub repository is provided, however there are no setup instructions, nor details such as the requirements and Python version. To our understanding, the code is for inference only.

## C Highly efficient modeling and optimization of neural fiber responses to electrical stimulation

**Highly efficient modeling and optimization of neural fiber responses to electrical stimulation**

**Link:** https://www.nature.com/articles/s41467-024-51709-8

**Application Field:** Computational neuroscience, modelling nerve fibre responses to electrical stimulation.

**Surrogate Focus:** Fully recreate the dynamics of the McIntyre-Richardson-Grill (MRG) axon, which is a gold standard example of a biophysically realistic model of a peripheral nerve. The goal is to mimic the response of the nerve fibre to electrical stimulation, while reducing the computational costs to improve the speed of optimising stimulation settings for designing protocols for inducing selective stimulation on peripheral nerves.

**Motivation for using a surrogate model**

- ✔ **Computationally expensive simulator:** The MRG axon is biophysically realistic, which means highly non-linear properties are simulated, inherently limiting the model with high computational costs. The state-of-the-art software for running the original MRG axon is the NEURON simulator, which is CPU-based and cannot run on GPUs, creating computational bottlenecks.
- ✔ **Computational gains and losses:** The paper reports the time taken to calculate the activation thresholds for a large number of fibres and stimulation protocols. These results indicate the surrogate model itself introduces a speedup of 2,000 to 130,000× speedup over single-core simulations in NEURON—all while maintaining a high accuracy. Additionally, the deployment of the models on GPUs also provides a 5 orders-of-magnitude speedup.

### Data Collection

- ✔ **Modes of input data:** Input data: 2D array representing the extracellular potential at every node and timepoint. The output is a 3D array representing the states of the neuron (membrane potential and four gating variables) at every node and time point

- ✔ **Sampling Strategy and Motivation:** Strategy is motivated by the input data, and by how the extracellular potential is used to manipulate the stimulus (amplitude, puse duration, stimulus delay) to modulate the axon model. While the fibre geometry was simplified to achieve scalability, the surrogate preserves the dynamic behaviour at the nodes of Ranvier, including the full spatiotemporal response of the gating variables, transmembrane currents, and transmembrane potential. Variables are sampled from a uniform distribution.

- ✔ **Data Formats:** Input: 2D arrays (extracellular potentials); output: 3D arrays/tensor (spatiotemporal response of axon to the stimulus).

- ✔ **Fidelity Level:** High fidelity level. The model is trained on the highly accurate numerical McIntyre-Richardson-Grill (MRG) model of a peripheral axon.

- ✔ **Data Availability:** Available at Github under "Generate training and/or validation datasets".

### Model Class

- ✔ **Deterministic or Probabilistic?:** Deterministic.
- ✘ **Predictive Uncertainty:** N/A.
- ✔ **Parametric Model:** Yes, parametric model, specifically a reparameterised and simplified biophysical cable model
- ✘ **Conditioning variables:** Not implemented.
- ✔ **Domain-specific constraints:** The surrogate model must be able to capture the full spatiotemporal response of the axon fibre, including the membrane potential ($V_m$) and all gating variables ($m, h, p, s$) at every node of Ranvier and time point. Further, it needs to account for the different stimulus parameters: waveform shape (monophasic, biphasic, arbitrary, etc.), the amplitude, frequency and the electrode configuration used to deliver the stimulus.

### Optimisation-based Learning

- ✔ **Optimiser and loss function:** Adam or RMSprop optimiser and mean squared error loss function.
- ✔ **Training Schedule:** Learning rate initialised to $1 \times 10^{-5}$. There are 5 epochs of training. The dataset consisted of 65,536 field-response pairs.
- **?** **Regularisation and early stopping criteria:** Weight decay is used as regularisation, and there is no explicit mention of whether early stopping is used.
- ✔ **Code availability and usability:** Available at AxonML. The README details the hardware and software requirements, installation methods and the steps for training the model, running simulations (pre-trained model is provided).

## D   Deep adaptive sampling for surrogate modelling without labeled data

### Deep Adaptive Sampling for Surrogate Modelling Without Labelled Data

**Link:** `https://link.springer.com/article/10.1007/s10915-024-02711-1`

**Application field:** Parametric differential equations in uncertainty quantification, inverse design, Bayesian inverse problems, digital twins, optimal control, shape optimisation.

#### Motivation for using a surrogate model

✔ **Computationally expensive numerical methods:** Repeatedly solving parametric differential equations is a computationally demanding task.

✔ **Computational gains and losses:** The surrogates with deep adaptive sampling provided speed ups of order exceeding $10^4$, when compared to classical solvers on problems of physics-informed operator learning, the parametric optimal control with geometrical parametrisation, and the parametric lid-driven 2D cavity flow with a continuous range of Reynolds numbers from 100 to 3200.

#### Data Collection

✔ **Modes of input data:** This work operates in a setting where pre-collected real world or pre-computed simulated "labelled data" is scarce as it relies on a physics informed approach where the surrogate model is trained by minimising a loss function that is based on the residuals of the parametric differential equations and boundary conditions.

✔ **Sampling strategy and motivation:** An adaptive approach uses a deep generative model (KRnet) to approximate a residual-induced distribution, iteratively refining samples in high-error regions. The authors motivate its use as it is core to the methods presented.

? **Data formats:** Input data format: Tuples $(x, \xi)$ combining spatial variables $x \in \Omega_s$ and parametric variables $\xi \in \Omega_p$; Output format: Scalar or vector solutions of differential equations (*e.g.* velocity, pressure in lid-driven cavity flow). The formats are not explicitly specified, nor indications are made if the standardised formats are used.

✔ **Data Availability:** Data is available on a GitHub repository.

## Model Class

- ✔ **Deterministic or probabilistic?:** Deterministic.
- ✘ **Predictive uncertainty:** N/A.
- ✔ **Parametric and Non-parametric:** Parametric, neural networks (feedforward or DeepONet) with 5–6 hidden layers of 20–50 neurons each.
- ✔ **Conditioning variables:** Two types of conditional PDF models are suggested but not utilised.
- ✔ **Domain-specific constraints:** Yes, this is a physics-informed network (loss function) and the domain constraints are implemented in terms of the governing parametric differential equations and boundary conditions.

## Optimisation-based learning: Physics informed network

- ✔ **Optimiser and loss function:** For the one-dimensional parametric ODE and operator learning problems, the Adam optimiser is used for all training. However, in the last two test cases, the optimal control problem and the parametric lid-driven cavity flow problems, the surrogate model is trained with the Broyden–Fletcher–Goldfarb–Shanno algorithm whilst the KRnet still uses Adam. Monte Carlo approximation of the loss functional is used as a training objective.
- ✔ **Training schedule:** Learning rate for the Adam optimiser is set to $1 \times 10^{-4}$, and the batch size is set to $10^3$.
- ✘ **Regularisation and early stopping criteria:** Not indicated.
- **?** **Code availability and usability:** Code is available on a GitHub repository, with set up instructions. While the dependencies are listed, this is done without specific versions.

## Diagnostics and Evaluation

- ✔ **Predictive performance:** The authors report relative $\ell_2$-error and time needed to perform inference.
- ✔ **Downstream task performance:** The model was compared to classical solvers and surrogates alternative sampling strategies in terms of the accuracy and time needed to perform inference physics-informed operator learning problem, the parametric optimal control problem with geometrical parametrisation, and the parametric lid-driven 2D cavity flow problem with a continuous range of Reynolds numbers from 100 to 3200.

