# OpenReview forum: "SMRS: advocating a unified reporting standard for surrogate models in the artificial intelligence era."
_NeurIPS.cc/2025/Position_Paper_Track — NeurIPS 2025 Position Paper Track_

### Official Review · Reviewer_iSsD · 2025-08-03

**Significance:** 2
**Presentation:** 3
**Rating:** 7
**Confidence:** 3

**Summary:**

This paper proposes the Surrogate Model Reporting Specification (SMRS), a unified and modular reporting standard designed to improve transparency, reproducibility, and cross-domain collaboration in surrogate modeling. Surrogate models, which approximate expensive simulations or physical processes, are increasingly used across scientific and engineering domains but suffer from inconsistent documentation of key decisions such as data sampling, model choice, uncertainty handling, and evaluation. SMRS outlines a structured schema covering these aspects and demonstrates its utility through case studies. While the framework is tailored to surrogate modeling’s unique needs—such as replacing known simulators rather than modeling observational data—it is designed to be model-agnostic and flexible to accommodate diverse methodologies. The authors argue that adopting SMRS can reduce fragmentation in the field, foster methodological rigor, and accelerate scientific progress.

**Strengths:**

The paper offers a timely discussion about the lack of standardization in surrogate modeling and proposes a possible solution through the Surrogate Model Reporting Specification (SMRS). The proposed framework is a reasonable starting point for improving reporting consistency. The inclusion of multiple case studies, drawn from a review of 17 papers, helps to illustrate the practical challenges and potential use of the SMRS. Overall, the paper raises a relevant issue and contributes to ongoing conversations around reproducibility and documentation in this area.

**Weaknesses:**

While the paper provides a thorough and structured discussion of what should be reported in surrogate modeling, the practical necessity and expected impact of this level of documentation could be made clearer. The authors propose SMRS as a solution, but the paper would benefit from a deeper discussion of why such detailed reporting is essential — especially in cases where code and data are already open-sourced. It remains somewhat unclear how SMRS would materially improve reproducibility, reuse, or model assessment beyond existing practices. A stronger justification of its added value, perhaps with concrete examples or user perspectives, would strengthen the case.

**Questions:**

While the paper proposes a detailed reporting framework for surrogate modeling, I’m unclear on what specific benefits this would offer beyond open-sourcing code and data. Could the authors clarify how SMRS adds value in scenarios where full implementation code is already shared? In particular, how does structured reporting improve reproducibility or model reuse in ways that well-documented repositories or notebooks might not?

**Alternative Position:**

Yes, and alternative positions are well-considered and addressed by the argument

**Author Identification:**

No.

**Context:**

3

**Discussion:**

3

**Ethics:**

["NO or VERY MINOR ethics concerns only"]

**Position:**

Yes, the paper argues for or against a position related to machine learning.

**Support:**

3

**Thoroughness:**

3

---

### Official Review · Reviewer_WKgk · 2025-08-08

**Significance:** 4
**Presentation:** 4
**Rating:** 7
**Confidence:** 4

**Summary:**

The paper advocates for a unified reporting standard, SMRS, for surrogate models across domains. It argues that fragmentation in current practice hampers reproducibility, comparability, and cross-domain knowledge transfer, and proposes a modular, model-agnostic schema covering data collection (including sampling design, fidelity), model class specification, learning methods, evaluation (diagnostic and task-based), and benchmarks. The authors discuss alternative viewpoints (e.g., risks of over-standardization, domain specificity) and address them. The case studies illustrate how the framework might be applied to existing and future literature.

**Strengths:**

* The paper makes a persuasive case that a unified reporting schema could significantly improve reproducibility, transparency, and cross-domain reuse of surrogate models. The argument for cross-domain knowledge transfer was convincing to me even though I was initially skeptical due to highly domain-specific practices.
* Data discussion was thorough, well structured by splitting into observational and simulated data. Appreciate the discussion of data selection, fidelity levels, active learning, etc. which will become more important as the field progresses, as well as the emphasis on data sampling which is often not reported or justified in papers.
* The deterministic vs probabilistic discussion, learning methods, sections are well reasoned and show an awareness of both classical and modern AI-based approaches.
* Alternative views section addressed genuine concerns.
* The call for benchmarks is important and well argued, with a reasonable scope that could realistically be adopted incrementally.
*  The case studies were helpful in showing the schema’s immediate use.

**Weaknesses:**

* While the paper discusses fidelity levels, it does not directly address surrogates trained from multiple heterogeneous simulation datasets (e.g., ClimateSet [1]). This might be a valuable scenario SMRS to address.
* Some applications mentioned (e.g., drug development) do not map cleanly to the fidelity concept, and the paper does not fully explain how SMRS should be adapted in such cases. This gap could make adoption harder in certain fields. Including some of these in the case studies would be helpful.
* The evaluation section could give more emphasis to defining good diagnostic and especially task-based evaluation protocols. Examples include metrics for extremes or quantiles, assessing scaling relative to naive baselines, and measuring the spread of predictive distributions for generative models. These could be explicitly encouraged in the schema rather than implied (an "X" mark for missing evaluation which goes beyond basic MSE, etc.).
*

**Questions:**

1. How would SMRS handle a surrogate trained on multiple simulation sources with differing definitions of fidelity?
2. Can the authors clarify how richer task-based evaluation metrics could be systematically incorporated into the schema to encourage their use?
3. For domains without a clear fidelity hierarchy (e.g. drug development), how should that part of the schema be adapted or marked as “not applicable”?
4. From the case studies, it seems the framework is especially useful when an outsider is able to evaluate the work and note deficiencies. An author self-reporting might write things more generously. Is there a way to address this? Further, how should the framework be used and by whom (in a new paper itself, as part of their documentation, in a survey work or benchmark work, etc.)? Some more discussion on the author's vision for implementation and use would be helpful.

**Alternative Position:**

Yes, and alternative positions are well-considered and addressed by the argument

**Author Identification:**

No.

**Context:**

3

**Discussion:**

3

**Ethics:**

["NO or VERY MINOR ethics concerns only"]

**Position:**

Yes, the paper argues for or against a position related to machine learning.

**Support:**

4

**Thoroughness:**

5

---

### Official Review · Reviewer_yEaf · 2025-08-23

**Significance:** 1
**Presentation:** 2
**Rating:** 2
**Confidence:** 5

**Summary:**

The paper argues for a position to use rigorous reporting schema when publishing new surrogate models.

**Strengths:**

1. The paper clearly argues in favor of using rigorous reporting schema when proposing new surrogate models.
2. The authors provide 3 case studies to support their arguments.

**Weaknesses:**

1. The argument of the paper is trivial. Any paper of any research field should use a rigorous reporting schema describing its computational gains and losses, data collection, model class, learning and evaluation methods.
2. The case studies are very limited and at times are wrongly cited: for example, the paper in case study in Section B uses the learning rate of 1e-3 and not 1e-4. The same optimizer is called Adam in Section B and ADAM in Section C indicating that the authors potentially simply copy-pasted arbitrary details from 3 papers, which they call case studies.

**Questions:**

1. I would be interested to understand, which information would authors like to convey with this paper apart from a trivial statement that any paper of any research field should use rigorous reporting schema?

**Alternative Position:**

Yes, and alternative positions are trivial straw-man arguments

**Author Identification:**

No.

**Context:**

2

**Discussion:**

1

**Ethics:**

["NO or VERY MINOR ethics concerns only"]

**Position:**

Yes, the paper argues for or against a position related to machine learning.

**Support:**

2

**Thoroughness:**

5

---

### Note · Authors · 2025-08-29

**1-10 Additional Comments:**

It is really frustrating not to be able to debate with reviewers, especially when their comments are dismissive.

**1-11 Submit Again:**

Unsure

**1-1 Submission Process:**

4

**1-2 Next Year:**

Clear selection criteria and an ability to write proper rebuttals and discuss with reviewers the way  it is done for the main track. More clarity on timelines.

**1-4 Interest:**

["Panel discussions with other position paper authors", "Structured debates on controversial topics", "Workshops for developing position papers", "Mentorship programs for early-career researchers"]

**1-5 Thoughtful:**

7

**1-7 Technical Aspects Versus Position:**

7

**1-8 Gate Keeping:**

4

**1-9 Camera Ready Changes:**

In the camera-ready version, we will
(i) correct minor inconsistencies in the current case studies shown in Appendix and expand their number to provide a broader and more rigorous demonstration of SMRS across domains;
(ii) clarify the value of SMRS beyond open-sourcing code/data;
(iii) add discussion on heterogeneous data sources and domains without a clear fidelity hierarchy;
(iv) enrich the evaluation section with examples of task-based metrics;
(v) give clearer guidance on how SMRS can be used by authors, reviewers, and benchmark creators.

**3-1 Review Response1:**

yEaf

**3-2 Reaction To Review1:**

Weakness 1: This reviewer finds our work trivial. If we had a chance for a proper rebuttal, we would invite them to fill out the SMRS form for a few papers; they would see that the task is far from trivial; moreover, they will find, as we did, that many of the required details are missing from papers in this field.


Weakness 2: The details we include are precisely those that must be documented using our proposed standardisation, and so are not “arbitrary”. We will re-check the learning rate; it looks like a typo.

The details the reviewer has highlighted are quite minor criticisms. We welcome disagreement on substance; but find the chosen tone disrespectful.

Our “case studies” are an implementation of our proposed standard on papers that we have not authored, and therefore would argue that they are by definition a case-study.



Question: We strongly agree that the criteria in SMRS should be a prerequisite for any paper using emulators. However, it is evident that most papers do not include this information.


Following the reviewer’s argument, one could equally argue that frameworks like model cards are trivial, since they report information that any paper in any research field should be providing anyway. Yet the paper introducing this idea is well-regarded, has nearly 3000 citations and has been widely implemented [Mitchell, Margaret, et al. "Model cards for model reporting." Proceedings of the conference on fairness, accountability, and transparency. 2019.] This makes the point that providing a rigorous framework to report essential information is important and has a beneficial impact on the field.

Should every paper be doing this? Yes! Are they doing this? Unfortunately, no!
Without a clear standard such as the one which we propose, research papers would only be reporting the steps which they undertook, and not reporting on the steps which they did not undertake.

**3-3 Review Response2:**

WKgk

**3-4 Reaction To Review2:**

We thank the reviewer for their very thoughtful comments.

Q1:
Great idea! We will add a multi-source fidelity section that, for each simulator, records values on shared fidelity axes, includes a small mapping table translating the simulator’s native labels to those axes with comparability marked as comparable/partial/not, and mandates stratified evaluation (incl. leave-one-source-out) with documented harmonisation steps.


Q2:
We’ll add a small “task-based evaluation” block: authors pick the task family, report at least two suitable metrics, include a baseline and compute budget, run one stress test (e.g., OOD/extremes/long-horizon), and note seeds and code path. If something doesn’t apply, they mark “N/A” with a one-line reason.

Q3:
In our current vision, this would be marked  “N/A”;

Q4:
This is an excellent point, thank you so much for raising this! SMRS is intended for both authors, reviewers, and benchmarking.

To limit self-reporting bias by authors, each field asks for an evidence link or “Not reported” with a one-line reason, and we could include a small completeness table summarising evidenced vs asserted vs missing items. Authors are encouraged to add a one-page SMRS card in the appendix as well as a repository, and pre-specify task metrics and seeds.

Reviewers can use a condensed checklist to spot omissions quickly and request fixes. Survey and benchmark papers publish third-party audited SMRS cards and can accept corrections via pull requests, enabling like-for-like comparisons on the same task metrics.

Our SMRS card could be an excellent tool for benchmarking. E.g. create a card for the benchmark, and for each of the subsequent models comparing themselves to the benchmark, it becomes very easy to track what's different using SMRS.

Other researchers: if one wants to build their research on an existing surrogate, the easiest place to start would be the SMRS card, and an attempt to undertake steps which were omitted in previous training.

**3-5 Review Response3:**

iSsD

**3-6 Reaction To Review3:**

“While the paper proposes a detailed reporting framework for surrogate modeling, I’m unclear on what specific benefits this would offer beyond open-sourcing code and data. Could the authors clarify how SMRS adds value in scenarios where full implementation code is already shared? In particular, how does structured reporting improve reproducibility or model reuse in ways that well-documented repositories or notebooks might not?”

We thank the reviewer for this question. Open-sourcing code and data addresses part of reproducibility, but in practice, repositories seldom provide a concise, declarative statement of the model’s scope and assumptions (e.g., validity ranges, simulator/version, boundary conditions), nor do they record what was not used (e.g., absence of a sampling strategy, no uncertainty calibration). These omissions matter for reuse and fair comparison.

SMRS adds a structured reporting card that
(i) makes data and sampling design explicit (offline/online, AL/BO criteria, splits), including negative reporting when a component is absent;
(ii) clarifies fidelity levels and heterogeneous sources and how they were reconciled;
(iii) states deterministic vs probabilistic outputs and the UQ assumptions with calibration evidence (e.g., SBC, coverage), rather than only point metrics;
(iv) specifies task-based evaluation protocols (extremes/quantiles, regret, calibration), enabling cross-paper comparability;
(v) centralises traceability (commit hashes, seeds, hardware) that is typically scattered across notebooks.
Well-documented repos still optimise for running code; SMRS complements them with a declarative specification that supports auditing, comparison, and reuse; even more so when particular elements are missing.

---

### Meta-Review · Area_Chair_4KCL · 2025-09-12

**Rating:** 6
**Confidence:** 4

**Strengths:**

The paper makes a case for the Surrogate Model Reporting Specification (SMRS), showing how a unified schema could improve reproducibility, transparency, and cross-domain reuse of surrogate models. While the idea of reporting standards might sound simple, the contribution is not trivial: the paper demonstrates that fragmentation across domains is a real practical barrier and that creating a flexible yet structured schema is challenging but impactful. The discussion of data issues (observational vs. simulated, fidelity, sampling), model choices (deterministic vs. probabilistic, uncertainty quantification), and evaluation practices is thorough and well grounded. Case studies effectively illustrate how SMRS can be applied in practice, and the paper also addresses genuine concerns about over-standardization.

**Weaknesses:**

The paper could better justify why structured reporting adds value beyond open-sourcing code and data, and it provides limited discussion of adoption challenges or incentives. The evaluation section, though covering diagnostics and task-based evaluation, could place stronger emphasis on richer metrics (e.g., quantiles, extremes, calibration of predictive distributions).

**Questions:**

Why is SMRS necessary when many papers already open-source code and data? What specific benefits does structured reporting provide for reproducibility, reuse, or comparability that repositories and notebooks do not?

In areas like drug development, where “fidelity levels” are unclear, how should the schema adapt? Should those sections be marked “N/A,” or is there a more nuanced way?

What strategies or incentives will make researchers across domains adopt SMRS? How will it integrate with existing tools, journals, or conference checklists without creating excessive burden?

**Ethics:**

There are no ethical concerns.

**Thoroughness:**

3

---

### Decision · Program_Chairs · 2025-09-26

Accept